# Physiological and Phytochemical Responses of Spinach Baby Leaves Grown in a PFAL System with LEDs and Saline Nutrient Solution

**Filippos Bantis [1],\*, Mariangela Fotelli [2] , Zoran S. Ilić [3] and Athanasios Koukounaras [1]**

[1] Department of Horticulture, Aristotle University, 54124 Thessaloniki, Greece; thankou@agro.auth.gr
[2] Forest Research Institute of Thessaloniki, Greek Agricultural Organization-Dimitra, Vassilika, 57006 Thessaloniki, Greece; fotelli@fri.gr
[3] Faculty of Agriculture, University of Priština-Kosovska Mitrovica, 38219 Lešak, Serbia; zoran.ilic63@gmail.com
**\*** Correspondence: fbantis@agro.auth.gr; Tel.: +30-2310-994123

**Abstract:** Spinach is a leafy vegetable containing a plethora of bioactive compounds. Our study aimed to evaluate the physiological (i.e., JIP-test) and phytochemical response of spinach baby leaves grown with regular or mildly saline (40 mM NaCl) nutrient solution and irradiated by four light-emitting diodes (LEDs) with broad spectra. T1 (highest red and far-red, low blue) and T3 (high red, balanced blue, green and far-red) led to a better developed photosynthetic apparatus compared to T2 (red peak in 631 nm) and T4 (highest blue and green), highlighted by $PI_{ABS}$ and its structural components: RC/ABS, $\varphi_{P0}$, $\psi_{E0}$, and $\Delta V_{IP}$. Elevated salinity only affected the latter parameter. T1 induced the maximum yield production but also the highest nitrate content which was far below the maximum level permitted by European legislation. Regardless of salinity level, T3 enhanced total phenol, chlorophyll, and carotenoid content. T2 and T4 led to inferior nutritional quality. Non-saline nutrient solution promoted the chlorophyll and carotenoid contents and the antioxidant potential, regardless of light treatment. By contrast, soluble sugar content was enhanced by saline nutrient solution. Our study shows that physiology and nutritional quality of spinach baby leaves can be manipulated by small interplays in the light spectra and salinity level.

**Keywords:** *Spinacia oleracea*; sodium chloride; light-emitting diodes; vertical farming; controlled environment agriculture; JIP-test; antioxidant potential; phenols; carotenoids; nitrates

---

## 1. Introduction

An active immune system is important for the human defense against chronic diseases and viral organisms such as the newly identified SARS-CoV-2. Naturally existing compounds have the potential to boost the immune system since they exhibit antiviral activity [1]; thus, the food sector is in the spotlight. Vegetables are essential for a healthy diet because they offer a plethora of bioactive compounds such as vitamins, phenolics, and carotenoids among other antioxidants providing a means of controlling oxidative stress in humans [2]. Especially baby leaf vegetables are particularly popular since they have a higher bioactive compound concentration compared to the respective fully grown plants [3]. The floating system is ideal for baby leaf vegetable production due to several benefits such as reduced cost, handling simplicity, efficient use of water and nutrients, absence of weeds and soil-borne pathogens, mechanical and automated harvesting, as well as greater yield and product quality. Since nutrient composition is customizable, this system offers the option to control bioactive compound accumulation in plants [4].

Spinach (*Spinacia oleracea* L.) is a worldwide cultivated, cool-season, leafy vegetable. It is rich in bioactive compounds that exhibit reactive-oxygen species scavenging activities and modulate expression of genes involved in human metabolism, inflammation, proliferation, and antioxidant defense among other benefits. However, spinach consumption is relatively low compared to other leafy vegetables [5]. Spinach struggles to thrive in summer temperatures [6] and climate change scenarios are not optimistic for the crop's development throughout a year. If we also take into consideration ground water salinity, spinach production is expected to deteriorate over the coming years [7]. High salinity in the nutrient solution can deteriorate crop productivity in non-halophytes since it triggers physiological responses involving ion accumulation in the vacuoles and solutes concentration in the cytoplasm in order to balance the cell water potential [8]. However, application of a mild salinity stress can trigger beneficial changes in plant metabolism involving their nutritional quality such as antioxidant potential [9].

Climate constraints can be weathered with protected farming in innovative plant factories with artificial lighting (PFALs). PFALs are nowadays gaining a lot of attention by researchers, growers, and stakeholders due to several important benefits. Specifically, PFALs can be established almost anywhere with emphasis in urban and periurban spaces in order to reduce "food miles", while plant growth needs close to zero agrochemical input. As a result, the products have higher safety and increased marketable period [10]. Nowadays, light-emitting diodes (LEDs) have several advantages for indoors cultivation compared to traditional light sources such as fluorescent (FL) lamps. Specifically, LEDs provide wavelength specificity, high light intensity, high energy conversion efficiency, long lifetime, and low thermal emission, and thus they are ideal as the sole light source for PFALs [11]. Moreover, product quality including antioxidant content can be manipulated by controlling light parameters such as quantity, quality, and duration [12].

Chlorophyll a fluorescence emitted by photosynthetic organisms upon illumination provides a lot of information about the photosynthetic apparatus. The JIP-test is an analysis of the efficiency of electron transport inside the intersystem chain from photosystem II (PSII) to the end electron acceptors at PSI side [13]. To this end, our study focused on the effect of broad-spectra light quality on the physiological status (i.e., chlorophyll fluorescence) of baby leaf spinach with the use of LEDs. Moreover, another aim was to test the potential of PFAL along with rather mild salinity stress to support baby leaf spinach cultivation with a goal of enhanced yield and nutritional quality.

## 2. Materials and Methods

### 2.1. Plant Material and Growth Conditions

The experiment was conducted at Forest Research Institute, Vassilika, Greece, and determinations were performed at the University farm. Seeds of *Spinacia oleraciea* L. cv. "Geant D' Hiver" were purchased from a local commercial store, sowed in polystyrene mini-plug trays (300 mm × 435 mm, 1.149 seedlings/m$^2$; 30 cc) filled with peat, and stored in darkness until germination. In order to ensure that all mini-plug cells contained a seedling, two seeds were sown per cell and were later thinned upon complete germination.

When full seedling emergence was achieved, the mini-plug trays were placed to float on top of water in polystyrene tanks (320 mm × 470 mm) filled with 16 L freshly produced Hoagland nutrient solution [100% strength; pH 6.5; electric conductivity (EC) 2.6 mS cm$^{-1}$]. Six tanks were placed under each light treatment in a growth chamber (shelf dimensions: 0.6 m × 1.2 m × 0.55 m, distance of 0.4 m between lamps and plants) where climate conditions were controlled at day/night temperature of 22 °C/20 °C, 80% ± 10% relative humidity, air was recirculating with fans, and light was applied with LEDs. Half of the tanks also included 40 mM sodium chloride (NaCl) in the nutrient solution (pH 6.5; EC 6.5 mS cm$^{-1}$). The specific NaCl concentration (40 mM) was selected due to its usual presence in saline water, as well as due to noteworthy results in other experiments of our group. The plants remained in the growth chamber for 28 days.

## 2.2. Light Conditions

Artificial lighting was emitted by four broad-spectra LEDs (120 cm, 132 W) emitting light with varying far-red, red, green, blue, and ultraviolet radiation percentages, hereby labeled as T1, T2, T3, and T4. Light distribution, photosynthetic photon flux density, and other spectral parameters (HD 30.1 spectroradiometer, DeltaOhm Srl, Padova, Italy) are presented in Table 1. Light treatments including saline water are indicated as T1+NaCl, T2+NaCl, T3+NaCl, and T4+NaCl, respectively. Photoperiod was set at 16 h, photosynthetic photon flux density (PPFD) was $150 \pm 10$ μmol m$^{-2}$ s$^{-1}$, while light was emitted by two lamps per light treatment. The particular lamps were selected due to the relatively small but important differences in the light spectra among each other according to a previous literature review. It should be noted that T2, T3, and T4 have a high color rendering index (CRI) (>50 units) constituting their light pleasant for personnel, as well as facilitating visual examination of the plants, in contrast to T1.

**Table 1.** Spectral distribution, red and blue peak wavelengths, red-to-blue (R:B) ratio, red-to-far red (R:FR) ratio, correlated color temperature (CCT), and color rendering index (CRI) of tested light treatments.

| Parameters | Light Treatment | | | |
|---|---|---|---|---|
| | T1 | T2 | T3 | T4 |
| UV%; 380–399 nm | 0.02 | 0.02 | 0.02 | 0.36 |
| Blue%; 400–499 nm | 7.62 | 10.90 | 11.38 | 20.59 |
| Green%; 500–599 nm | 2.34 | 18.54 | 13.85 | 36.46 |
| Red%; 600–699 nm | 67.25 | 62.20 | 56.48 | 36.92 |
| Far-red%; 700–780 nm | 22.77 | 8.34 | 18.28 | 5.68 |
| Red peak (nm) | 660 | 631 | 660 | 660 |
| Blue peak (nm) | 448 | 448 | 448 | 461 |
| R:B | 8.82 | 5.71 | 4.97 | 1.79 |
| R:FR | 2.95 | 7.46 | 3.09 | 6.50 |
| CCT (K) | - | 1624 | 2143 | 5034 |
| CRI | - | 66.1 | 71.0 | 87.7 |

## 2.3. Yield, Color, and Chlorophyll Fluorescence

Following 28 days of growth in the chamber, color and chlorophyll fluorescence (JIP-test) were determined, and afterwards all plants were harvested in order to determine the leaf mass per area (i.e., yield). Colorimetric data (lightness, a* and b* coordinates) were obtained with a colorimeter (CR-400 Chroma Meter, Konica Minolta Inc., Tokyo, Japan).

Chlorophyll fluorescence measurements were performed at night on the first fully developed leaf of 10 plants per light treatment and salinity level. A pocket plant efficiency analyzer (PEA) chlorophyll fluorometer (Hansatech, King's Lynn, UK) was used to determine the induction curve of chlorophyll fluorescence emitted by photosystem II (PSII) during the first second of dark-adapted leaves' illumination (OJIP transient). PEA Plus 1.0.0.1 (Hansatech) and Biolyzer 4HP (Bioenergetics Lab., Univ. of Geneva, Geneva, Switzerland) were used for data analysis. In this study, we present and discuss results from parameters such as $PI_{ABS}$ (performance index), RC/ABS ($Q_A$ reducing reaction centers per PSII antenna), $\varphi_{P0}$ (maximum quantum yield for primary photochemistry), $\psi_{E0}$ (probability that an electron moves further than $Q_A$), and $\Delta V_{IP}$ (relative fluorescence increase between the intersystem carriers and electron end acceptors of PSI) (cf. [13]).

## 2.4. Phytochemical Composition

The harvested plant material was stored at −30 °C for about a week. Afterwards, spinach leaves from each treatment were macerated in a blender. Soluble sugar content (SSC) was directly determined from the filtered extract using a refractometer (PAL-α, Atago, Tokyo, Japan) and expressed as g of sucrose per 100 g of solution. For total phenolic and antioxidant capacity determination,

samples (2.5 g) were extracted into 25 mL 80% aqueous methanol. Total phenols were determined according to the method of Singleton and Rossi [14]. Briefly, 0.5 mL methanolic plant extract, 2.5 mL of Folin–Ciocalteau's reagent, and 2 mL of 7.5% sodium carbonate solution were incubated at 50 °C for 5 min. Absorbance of the colored product was measured at 760 nm, and the results were expressed as mg of gallic acid equivalent/g fresh weight.

Antioxidant capacity was determined by ferric reducing antioxidant power (FRAP) assay according to Benzie and Strain [15]. Briefly, 3 mL working solution ($CH_3COONa$ buffer solution, pH 3.6; TPTZ and $FeCl_3$) were added to 0.1 mL methanolic plant extract and incubated at 37 °C for 4 min. Absorbance of the colored product was measured at 593 nm and the results were expressed as μg of ascorbic acid equivalent/g fresh weight.

Nitrate content was determined according to Cataldo et al. [16]. Briefly, samples (2.5 g) were extracted into 25 mL water. An aliquot of 0.2 mL aqueous plant extract was added in 0.8 mL $H_2SO_4$ or 5% salicylic acid in $H_2SO_4$, which was followed by addition of 19 mL 2 N NaOH. Absorbance of the colored product was measured at 410 nm and the results were expressed as mg/kg fresh weight.

Chlorophyll a (*chl a*) and b (*chl b*), and carotenoid contents were determined according to Sumanta et al. [17]. Samples (0.5 g) were extracted into 10 mL 80% aqueous acetone and incubated in the dark for 24 h. After two centrifugations (10 min, 10,000 rpm, 4 °C), 80% aqueous acetone was added to a total of 100 mL. Absorptions were measured at 663, 647, and 470 nm. Concentrations of *chl a*, *chl b*, and carotenoids were calculated using the following equations:

$$Chl\ a = 12.25 \times A_{663.2} - 279 \times A_{646.8}$$

$$Chl\ b = 21.50 \times A_{646.8} - 5.10 \times A_{663.2}$$

$$Carotenoids = (1000 \times A_{470} - 1.82 \times Chl\ a - 85.02 \times Chl\ b)/198$$

*2.5. Statistical Analysis*

Each treatment had three biological replications (tanks with nutrient solution). Analysis of variances (ANOVA) was conducted by IBM SPSS software (SPSS 25.0, IBM Corp., Armonk, NY, USA). Comparisons of the means were executed with Tukey's test at significance level $\alpha = 0.05$. Comparisons between all non-saline (T1, T2, T3, and T4) and all saline (T1+NaCl, T2+NaCl, T3+NaCl, and T4+NaCl) treatments were conducted using *t*-test at $\alpha = 0.05$. Correlations between colorimetric parameters, and chlorophylls' and carotenoid content were tested using Pearson's correlation coefficient. Normality was tested with Kolmogorov–Smirnov test. The experiment was performed twice reaching similar conclusions, thus, results from the first repetition are presented.

**3. Results and Discussion**

Leaf mass per area was expressed as $g/m^2$ and was variable among the light and salinity treatments. Specifically, T1 and T1+NaCl (highest red and far-red, lowest blue) induced the production of significantly greater fresh biomass compared to T2 (highest red/far-red; +73% and +66%, respectively) and T4 (highest blue and green, high red/far-red; +118% and +108%, respectively) (Figure 1A), while no differences were detected among saline and non-saline treatments. Addition of far-red light can be associated with respective increase in shoot biomass and several morphological characteristics. Moreover, high amount of blue light is known to suppress extension growth (i.e., shoot weight production) via the cryptochrome photoreceptors [18]. For example, one basil and two lettuce cultivars produced significantly greater fresh biomass under the influence of 1-to-1 red–blue ratio including 30 μmol m$^{-2}$ s$^{-1}$ far-red light, compared to monochromatic red or blue LEDs and other red–blue combinations [19].

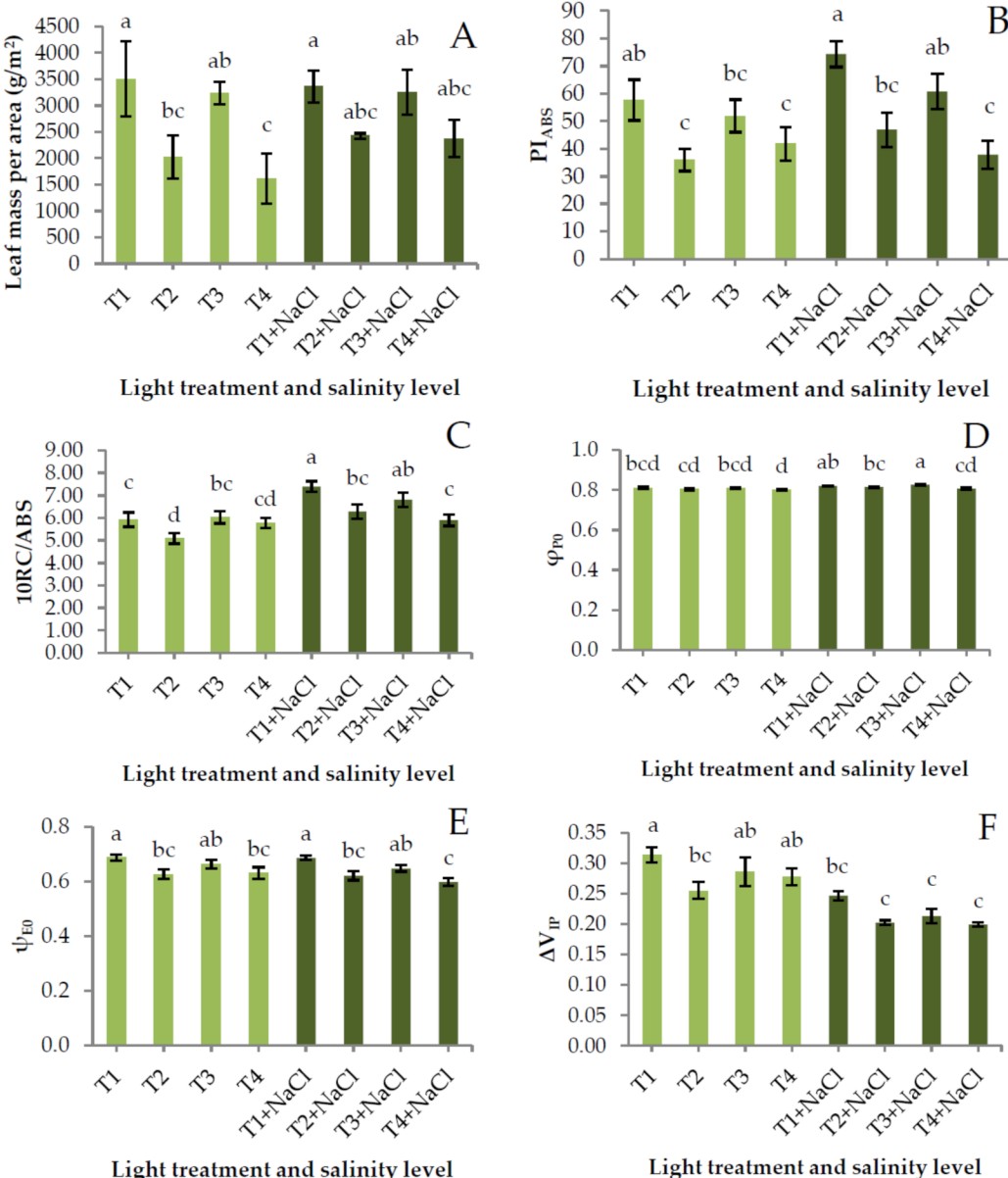

**Figure 1.** (**A**) Leaf mass per area, (**B**) performance index on absorption basis (PI$_{ABS}$), (**C**) density of active reaction centers (10RC/ABS), (**D**) quantum efficiency of reduction of Q$_A$ ($\varphi_{Po}$), (**E**) probability of electron transport beyond Q$_{A-}$ ($\Psi_{Eo}$), and (**F**) relative fluorescence increase between I- and P-step ($\Delta V_{IP}$) of baby leaf spinach cultivated in a growth chamber under four light treatments and two salinity levels. Bars (±SE) followed by different letters indicate significant differences ($p \leq 0.05$). Leaf mass per area values was computed from $n = 3$ tanks. JIP-test parameter values were compiled from $n = 10$ plants.

According to the relatively high values obtained from chlorophyll fluorescence measurements, plants developed well in all treatments. Monochromatic red light is known to be damaging for the photosynthetic apparatus and especially PSII [20]. Blue addition to red background light has the potential to ameliorate the abovementioned damaging effects [21]. In our study, PI$_{ABS}$, which summarizes the effects of RC/ABS, $\varphi_{P0}$, and $\psi_{E0}$, was significantly greater in T1+NaCl (highest red and far-red, low blue) followed by T1 and T3+NaCl (Figure 1B), while light trapping (RC/ABS) followed a similar trend (Figure 1C). These trends show that differences in PI$_{ABS}$ can be attributed mainly to RC/ABS. Lower ABS/RC (i.e., higher RC/ABS) was found in green and purple basil under 50% red-50% blue and 70% red-30% blue wavelengths, compared to monochromatic red light [22]. Quantum efficiency of Q$_A$ reduction ($\varphi_{P0}$) was significantly higher in T3+NaCl followed by T1+NaCl (Figure 1D),

while probability of electron transport from $Q_A$ to the intersystem carriers of the electron chain ($\psi_{E0}$) was enhanced in T1 and T1+NaCl, followed by T3 and T3+NaCl (Figure 1E). However, all treatments exhibited relatively high values in both parameters. For example, $\varphi_{P0}$ (i.e., Fv/Fm) reaches values in the range of 0.78–0.86 in non-stressed plants of 44 tested species [23]. These adjustments take place in order to control energy dissipation and reduce the overexcitation. Electron flow is reduced within the transport chain when photosynthesis is restricted and reductant potential cannot be used for metabolism [24]. Relative fluorescence increase between the intersystem carriers and electron end acceptors of PSI ($\Delta V_{IP}$) was promoted in T1, followed by T3 and T4, compared to all salinity treatments (Figure 1F). In general, t-tests showed that treatments with non-saline nutrient solution reached higher $\Delta V_{IP}$ values compared to salinity treatments ($p < 0.001$). Salinity negatively affects the electron transport chain, and this is highlighted by $\Delta V_{IP}$ values in our study. Salinity is related to drought which limits Rubisco ability to receive electrons from NADPH, thus blocking electrons at the PSI acceptor side due to reduced ferredoxin NADP+-reductase activity [25]. In general, treatments involving T2 (i.e., T2, T2+NaCl) and T4 (i.e., T4 and T4+NaCl) led to the lowest values in all studied parameters (Figure 1B–F). This shows that regardless of salinity level, plants grown under T2 and T4 were more stressed compared to T1 and T3. In our study, T2 emits relatively high amounts of red light with peak in 631 nm (orange-red). Ahlman et al. [26] reported slightly greater fluorescence gain (index of photosynthetic efficiency increase) in lettuce exposed to 160 μmol m$^{-2}$ s$^{-1}$ of 660 nm red light compared to 630 nm, while cucumber showed similar values. Moreover, strawberry plants showed similar photosynthesis under 630 and 660 nm red lights [27]. T4 light treatment emits a small portion of UV (<1%) as well as 21% blue light which, according to Planck's law ($E = h \times c \times \lambda^{-1}$), contain a significant amount of energy compared to red wavelengths, leading to PSII photodamage [28].

Accumulation of phytochemical compounds is a means of plant adaptation to biotic and abiotic environmental constraints. Phenolic compounds are particularly active in the defense and signaling mechanisms while they are also involved in neutralization of reactive oxygen species [29]. These compounds are accumulated in plant tissues in response to various environmental factors including light quantity and quality providing information about possible stressful growth conditions. In our study, total phenol content was significantly greater under T3+NaCl (high red with balanced blue, green, and far-red) compared to T2+NaCl (highest red/far-red; +64%) and T4+NaCl (highest blue and green, high red/far-red; +51%) (Figure 2A), while no significant differences were observed among treatments with or without additional salinity in the nutrient solution. By contrast, phenolic content of lettuce baby leaves was enhanced under saline conditions [30]. Lamb's lettuce (*Valerianella locusta*) developed greater amount of total phenols, as well as individual phenolic compounds (i.e., phenolic acids and flavonoids) under the influence of 70% red–30% blue light treatment compared to narrow-band red and other red–blue combinations [31]. In two studies of our group, basil seedlings accumulated more phenolics under increasing blue light composition (similar treatment to T4) [32], while *Salvia fruticosa* seedlings showed tendency for increased total phenols under a treatment similar to T3 of the present study but did not exhibit significant differences [33].

Chlorophylls are essential pigments which capture photons and use the produced energy for photosynthesis. Specifically, chlorophyll a captures light energy at 430 nm and 665 nm peak wavelengths, while chlorophyll b captures light energy at 453 nm and 642 nm peak wavelengths [34]. Therefore, the abovementioned red (600–700 nm) and blue (420–450 nm) wavelengths are critical for photosynthesis. In our case, the content of chlorophyll a was significantly elevated under T3 (high red with balanced blue, green, and far-red) compared to almost all treatments, apart from T2 (highest red/far-red) and T3+NaCl, while total chlorophyll content followed the same trend. Chlorophyll b content was also greater under T3 compared to T1+NaCl (highest red and far-red, lowest blue), T2+NaCl, and T4+NaCl (highest blue and green, high red/far-red) (Figure 2B). Nanzin et al. [35] reported that the 91%/9% red/blue composition led to the highest chlorophyll amount in spinach, lettuce, basil, and pepper compared to 83%/17%, 95%/5% and 100%/0%. It is clear that blue light has a critical role involving chlorophyll regulation since it is known to act through phototropin photoreceptors

(phot I and II) and regulates chloroplast movement in the cells [36]. As a general trend, a *t*-test showed that treatments with no additional salinity developed greater total chlorophyll content ($p = 0.001$), chlorophyll a ($p = 0.001$), and chlorophyll b ($p = 0.002$) contents compared to treatments with saline water. Similarly, *Cichorium spinosum* produced less chlorophylls a and b when treated with saline water compared to regular nutrient solution [37]. On the contrary, in another study with spinach, the authors reported higher chlorophyll and carotenoid contents under mild salt stress compared to no salinity in the nutrient solution [38]. Differences in the results can be attributed to application method, salinity strength, and other factors.

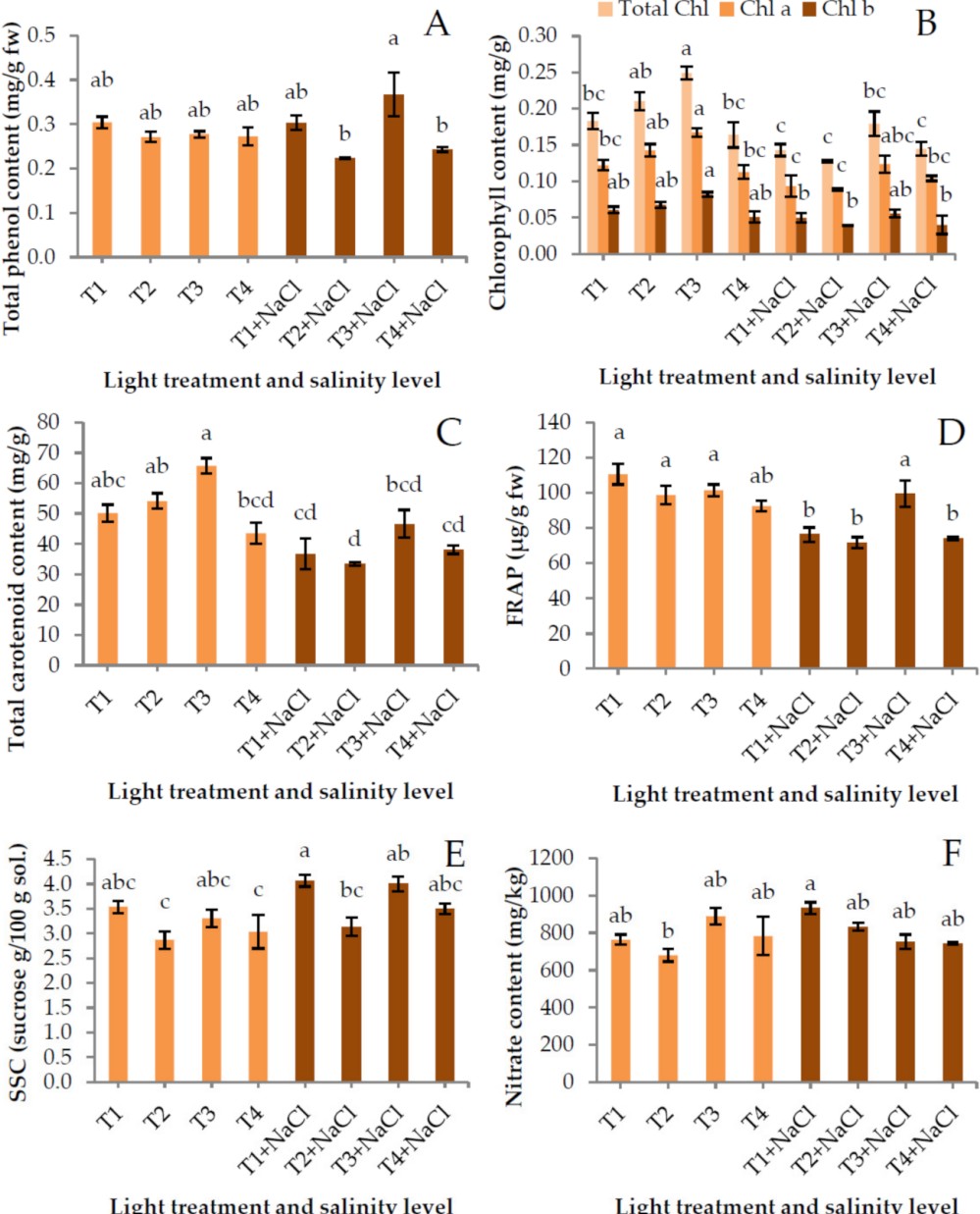

**Figure 2.** (**A**) Total phenol content, (**B**) total chlorophyll, chlorophyll a (*chl a*), and chlorophyll b (*chl b*) contents, (**C**) total carotenoid content, (**D**) ferric reducing antioxidant power (FRAP), (**E**) soluble sugar content (SSC), and (**F**) nitrate content of baby leaf spinach cultivated in a growth chamber under four light treatments and two salinity levels. Bars (±SE) followed by different letters indicate significant differences ($p \leq 0.05$). Mean values were compiled from $n = 3$ tanks.

Carotenoids are accessory pigments which harvest light and impose an important role in the transduction of photosynthetic energy [39]. In particular, carotenoids limit membrane damage and extend chlorophyll life through radical detoxification and excess energy dissipation [40]. Quite similarly to chlorophylls, T3 induced the production of greater total carotenoid content compared to T4 (+51%) and all salinity treatments (+41–96%) (Figure 2C). In general, a *t*-test showed that treatments with saline water exhibited significantly lower values compared to treatments with non-saline water ($p < 0.001$). Enhanced carotenoid production was reported in lettuce grown under red-blue LED, compared to white, red and blue LEDs [41]. In a more recent study, a treatment including 17% blue light (and 83% red) was beneficial for enhanced total carotenoid content of spinach, lettuce, kale, basil, and pepper compared to narrow-band red and red-blue combinations with higher amount of red light [35]. Moreover, two lettuce cultivars grown under supplementary blue light treatments showed enhanced production of carotenoids such as lutein, β-carotene, zeaxanthin, neoxanthin, and violaxanthin [42]. Similarly to chlorophyll, the presence of blue light in relatively high portions clearly promotes carotenoid production and accumulation in plants. Both pigment types (i.e., chlorophylls and carotenoids) absorb blue light which enhances photosynthesis. However, when the amount of blue light is extremely high, it may damage the photosynthetic apparatus and trigger the production of pigments to restore the efficiency of the system.

Antioxidant potential expressed as FRAP was significantly higher under T3+NaCl (+30–39%) compared to the rest of the salinity treatments (Figure 2D). Strawberry fruits exhibited significantly greater FRAP in light with a red–blue ratio of 1.5 and 5.5 compared to a red–blue ratio of 0.7 [43], while lamb's lettuce showed greater DPPH scavenging activity under 90% red–10% blue compared to 100% red and other red–blue combinations [44]. In our study, differences were less pronounced among light treatments without salinity. However, T1 and T2 treatments without salinity showed significantly higher FRAP values ($p < 0.001$) compared to the same treatments with additional salinity, as shown by t-test. In studies with other species, the effect of salinity on ascorbic acid level was not consistent. For example, in a study with red lettuce, plants treated with 20 mM NaCl produced more ascorbic acid compared to 1, 10, and 30 mM NaCl [8], while Hamilton and Fonseca [9] found greater ascorbic acid content in *Eruca sativa*, *Diplotaxis tenuifolia*, and *Lepidium sativum* treated with the least salt in the nutrient solution.

Soluble sugars are important compounds for evaluating the nutritional quality as well as the taste of leafy vegetables. In our study, soluble sugar content exhibited higher values in T1+NaCl (highest red and far-red, lowest blue) compared to T2 (+23%), T2+NaCl (highest red/far-red; +13%), and T4 (highest blue and green, high red/far-red; +16%) (Figure 2E). In a study with lettuce, greater soluble sugar content was found in lettuce treated with white-red LED compared to white-blue LED among other light treatments [45]. In a study with lamb's lettuce, plants treated with 90% red–10% blue (quite similar to our T1) produced more soluble sugars compared to narrow-band red and other red–blue combinations [44]. Moreover, a t-test showed that the application of additional salinity in our study resulted in significantly higher levels of soluble sugars compared to treatments with regular nutrient solution ($p = 0.011$). Quite similarly, in a study involving lettuce, soluble sugar content was greater in saline substrate with low water potential [46].

Nitrate ions are widely considered as detrimental for human health, since they are a source of nitrosamines which are carcinogenic compounds [47]. A high amount of nitrates (75%) are consumed through vegetables [48] such as spinach. It is well-established that light intensity is a critical environmental parameter that regulates nitrate concentration in leafy vegetables. However, light quality is also a major factor for nitrate control in plants [49]. Baby spinach leaves under T1+NaCl (highest red and far-red, lowest blue) produced a significantly greater amount of nitrates only compared to T2 (highest red/far-red; +37%) (Figure 2F). In all treatments, nitrate content was well below 2500 mg/kg which is the legal upper limit in the European Union. Chung et al. [46] found that nitrate concentration in lettuce increased up to EC of 5 dS m$^{-1}$ (i.e., with moderate salinity levels) but in our case no differences were observed between treatments with saline and non-saline nutrient solution. Several researchers studied the impact of light

wavelength on nitrate content of leafy vegetables and, contrasting to our results, they found a tendency for lower nitrate accumulation under increasing red light content [50]. In addition, nitrate accumulation is reportedly negatively correlated to soluble sugar content since sugars are a source of energy and carbon for the metabolism of nitrogen [51], and they have common osmoregulative functions with $NO_3$ ions [52]. No negative correlation was detected in our case indicating possible species or harvest-stage dependency.

Leaf color is a major visual quality index for leafy vegetables. In spinach, leaves with deep green color are associated with high-quality product and consumers tend to prefer them. Colorimetry revealed significant differences in three out of four tested parameters, except for lightness which was similar for all treatments (Table 2). Chroma and a*/b* coordinates were greater under T3+NaCl compared to T1 and T2, and T2, T3 and T4, respectively, while hue angle showed the opposite results than a*/b* coordinates (Table 2). Colorimetric parameters are usually correlated with chlorophyll and carotenoid contents [53]. However, in our study we did not observe any significant correlations between the abovementioned parameters. Specifically, chlorophyll a and b, total chlorophyll and total carotenoid content showed Pearson's r values between −0.56 and −0.37 compared to lightness, between −0.25 and −0.07 compared to chroma, between 0.25 and 0.41 compared to hue angle, and between −0.36 and −0.19 compared to a*/b* coordinates.

**Table 2.** Colorimetric parameters of baby leaf spinach cultivated in a growth chamber under four light treatments and two salinity levels.

| Light Treatment | Colorimetric Parameter | | | |
|---|---|---|---|---|
| | Lightness | Chroma | Hue Angle | a*/b* |
| T1 | 44.62 ± 0.58 [a] | 29.23 ± 0.91 [c] | 125.05 ± 0.37 [ab] | −0.70 ± 0.01 [ab] |
| T2 | 44.60 ± 0.42 [a] | 29.75 ± 0.67 [bc] | 125.79 ± 0.33 [a] | −0.72 ± 0.01 [b] |
| T3 | 46.21 ± 0.52 [a] | 30.73 ± 0.71 [abc] | 125.42 ± 0.36 [a] | −0.71 ± 0.01 [b] |
| T4 | 45.30 ± 0.57 [a] | 30.69 ± 0.96 [abc] | 125.32 ± 0.40 [a] | −0.71 ± 0.01 [b] |
| T1+NaCl | 45.75 ± 0.79 [a] | 34.21 ± 1.02 [ab] | 123.82 ± 0.41 [ab] | −0.67 ± 0.01 [ab] |
| T2+NaCl | 43.76 ± 0.76 [a] | 32.01 ± 1.24 [abc] | 125.20 ± 0.47 [ab] | −0.71 ± 0.01 [ab] |
| T3+NaCl | 45.74 ± 0.98 [a] | 34.72 ± 1.56 [a] | 123.25 ± 0.64 [b] | −0.66 ± 0.02 [a] |
| T4+NaCl | 43.62 ± 1.01 [a] | 32.88 ± 1.35 [abc] | 124.95 ± 0.59 [ab] | −0.70 ± 0.02 [ab] |

Within a column, mean values (± SE) followed by different letters indicate significant differences ($p \leq 0.05$). a*: red/green coordinate; b*: yellow/blue coordinate. Each mean value was compiled from $n$ = 15 plants.

## 4. Conclusions

Spinach production is greatly impacted by elevated temperatures and salinity, both imposed by anthropogenic activities. These environmental constraints can be weathered by plant factories with artificial light (PFALs) using LEDs with varying spectra, which may also improve the quality of produced spinach plants. To address these issues, we studied the physiological and phytochemical responses of spinach grown under varying light qualities and contrasting salinity. Even a small addition of salt (40 mM) in the nutrient solution led to inferior quality, especially in terms of chlorophylls, carotenoids, and antioxidant potential. Parameters of the JIP test (RC/ABS, $\varphi_{P0}$, $\psi_{E0}$, and $\Delta V_{IP}$, and $PI_{ABS}$) indicated that among all tested light spectra combinations, the photosynthetic apparatus was better developed when plants were treated with high proportion of red light, either with far-red and low blue (T1) or with balanced blue, green and far-red (T3). Furthermore, T1 enhanced yield production, while T3 improved the nutritional quality of the plants by inducing the production of phenols, chlorophylls, and carotenoids. Other light combinations with small differences in far-red, blue, or green spectra led to the production of plants with less desired traits. Our results emphasize that minor interplays in light spectra and salinity in the growth medium are powerful tools for promoting the production of spinach plants rich in metabolites that support a strong immune system. Future research on this topic would benefit from molecular analyses on genes involved in the physiological modifications and biochemical pathways in response to light quality.

**Author Contributions:** Conceptualization, methodology, and data analysis: F.B. and A.K.; experimental measurements: F.B. and M.F.; writing—original draft preparation: F.B.; writing—review and editing: F.B., A.K., Z.S.I., and M.F.; supervision and project administration: A.K. All authors have read and agreed to the published version of the manuscript.

**Funding:** This research received no external funding.

**Conflicts of Interest:** The authors declare no conflict of interest. The founding sponsors had no role in the design of the study, the collection, analyses, or interpretation of data, the writing of the manuscript or the decision to publish the results.

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
