# Peer review of "Physiological and Phytochemical Responses of Spinach Baby Leaves Grown in a PFAL System with LEDs and Saline Nutrient Solution"

_agriculture, doi:10.3390/agriculture10110574_

Round 1

Reviewer 1 Report

The manuscript “Physiological and phytochemical responses of spinach baby leaves grown in a PFAL system with LEDs and saline nutrient solution” by Bantis et al. is an interesting and well-written paper describing a study with important implications for leafy green production in protected production systems using artificial lighting. I have a few specific comments listed below.

Line 38 – The language: “they can easily be handled by consumers as well as producers” is somewhat confusing. Are you saying that consumers and producers (farmers) find fully-grown leafy greens difficult to handle? Producing, harvesting, washing, packaging, and distributing baby greens can actually be more challenging compared to mature plants because they're more fragile and have a shorter shelf life.

Line 41 – Among the list of specific benefits attributed to the floating system, the authors include “environmentally friendly.” This is overly broad, vague and debatable. Recommend deleting it.

Line 50 – The authors write: “If we also take into consideration ground water salinity, spinach production is expected to deteriorate over the coming years.” This seems like a pretty broad statement without justification. Do you mean globally or just in the Mediterranean region?

Line 73 – The authors write: “…to support baby leaf spinach cultivation with view to enhanced yield and nutritional quality.” Do you mean “…with a goal of enhanced…”?

Line 102 – Interesting note.

Line 106 – Color and colour seemed to be used interchangeably in the manuscript and, while they mean the same, it would be less distracting for readers to go with one spelling or the other throughout.

Line 282 – Can you provide the genus and species for lamb’s lettuce at first mention?

Author Response

Statement by authors: Parts in the manuscript modified in response to Reviewer #1 comments are highlighted with yellow colour.

The manuscript “Physiological and phytochemical responses of spinach baby leaves grown in a PFAL system with LEDs and saline nutrient solution” by Bantis et al. is an interesting and well-written paper describing a study with important implications for leafy green production in protected production systems using artificial lighting. I have a few specific comments listed below.

Line 38 – The language: “they can easily be handled by consumers as well as producers” is somewhat confusing. Are you saying that consumers and producers (farmers) find fully-grown leafy greens difficult to handle? Producing, harvesting, washing, packaging, and distributing baby greens can actually be more challenging compared to mature plants because they're more fragile and have a shorter shelf life.

  • Response, line 38: Our initial thought was to highlight the fact that baby greens are more compact and can easily be produced by growers (mainly due to shorter cultivation time) and handled by consumers (due to easy washing and quick salad preparation). However, this part was removed since it is confusing.

Line 41 – Among the list of specific benefits attributed to the floating system, the authors include “environmentally friendly.” This is overly broad, vague and debatable. Recommend deleting it.

  • Response, line 41: The term was deleted as you suggested.

Line 50 – The authors write: “If we also take into consideration ground water salinity, spinach production is expected to deteriorate over the coming years.” This seems like a pretty broad statement without justification. Do you mean globally or just in the Mediterranean region?

  • Response, line 52: A reference was added. According to Gholami et al. (2010), groundwater salinity of urban societies throughout the world has increased and is predicated that it will increase up to 60% in the year 2030.

Line 73 – The authors write: “…to support baby leaf spinach cultivation with view to enhanced yield and nutritional quality.” Do you mean “…with a goal of enhanced…”?

  • Response, line 74: Thank you for the observation. The sentence was amended accordingly.

Line 102 – Interesting note.

  • Response: Thank you for your comment.

Line 106 – Color and colour seemed to be used interchangeably in the manuscript and, while they mean the same, it would be less distracting for readers to go with one spelling or the other throughout.

  • Response: The word “colour” is now used throughout the text for consistency.

Line 282 – Can you provide the genus and species for lamb’s lettuce at first mention?

  • Response, line 219: Genus and species for lamb’s lettuce (Valerianella locusta) was added at first mention, as you suggested.

Reviewer 2 Report

Reviewed manuscript entitled “„Physiological and phytochemical responses of spinach baby leaves grown in a PFAL system with LEDs and saline nutrient solution” is an original research study.

Following are the detailed suggestion for the Authors:

What was the criterion for choosing these lamps? Was it confirmed by previous study?

Was the normal distribution of data evaluated using Shapiro-Wilk test?

line 315 and 326 – tables with correlation should be provided.

It would be better if you could ended your conclusion with one sentence regarding the vision of the future of your studies.

Author Response

Statement by authors: Parts in the manuscript modified in response to Reviewer #2 comments are highlighted with cyan colour.

Reviewed manuscript entitled “„Physiological and phytochemical responses of spinach baby leaves grown in a PFAL system with LEDs and saline nutrient solution” is an original research study.

Following are the detailed suggestion for the Authors:

What was the criterion for choosing these lamps? Was it confirmed by previous study?

  • Response: The particular lamps were selected due to the relatively small but important differences in the light spectra among each other. For example, percentage of red light is between 37 and 67% while blue light is 8-21%. Moreover, we believe that narrow-band red or blue lamps or bichromatic red-blue lamps should be considered for replacement with broad-spectra lamps which are friendlier for the human eyes and facilitate visual examination of the plants. It is specified in lines 105-106.

Previous studies of our group showed promising results for the production of seedlings from various species (trees, herbs, and vegetables) with the use of these lamps and thus we aimed to test them for the production of edible vegetables as well.

Was the normal distribution of data evaluated using Shapiro-Wilk test?

  • Response: Normal distribution of data was tested with Kolmogorov-Smirnov test. It is now specified in the lines 164-165.

line 315 and 326 – tables with correlation should be provided.

  • Response: Thank you for your comment. We did not observe any significant correlations between colorimetric parameters and chlorophylls’ and carotenoid content. However, we added Pearson’s r values in the text (cf. lines 335-338).

It would be better if you could ended your conclusion with one sentence regarding the vision of the future of your studies.

  • Response: Thank you for the comment. A concluding sentence was added involving possible future research activity, as you suggested (cf. lines 361-362).